# Breast cancer incidence in Yogyakarta, Indonesia from 2008–2019: A cross-sectional study using trend analysis and geographical information system

**Bryant Ng**[1], **Herindita Puspitaningtyas**[2], **Juan Adrian Wiranata**[3,4], **Susanna Hilda Hutajulu**[5]*, **Irianiwati Widodo**[6], **Nungki Anggorowati**[6], **Guardian Yoki Sanjaya**[7], **Lutfan Lazuardi**[7], **Patumrat Sripan**[8]

**1** Faculty of Medicine, Medicine Study Program, Public Health and Nursing, Universitas Gadjah Mada/Dr. Sardjito General Hospital, Yogyakarta, Indonesia, **2** Faculty of Medicine, Doctorate Program of Health and Medical Science, Public Health and Nursing, Universitas Gadjah Mada, Yogyakarta, Indonesia, **3** Academic Hospital, Universitas Gadjah Mada, Yogyakarta, Indonesia, **4** Faculty of Medicine, Master Program in Clinical Epidemiology, Public Health and Nursing, Universitas Gadjah Mada, Yogyakarta, Indonesia, **5** Faculty of Medicine, Department of Internal Medicine, Division of Hematology and Medical Oncology, Public Health and Nursing, Universitas Gadjah Mada/Dr. Sardjito General Hospital, Yogyakarta, Indonesia, **6** Faculty of Medicine, Department of Anatomical Pathology, Public Health and Nursing, Universitas Gadjah Mada, Yogyakarta, Indonesia, **7** Faculty of Medicine, Department of Health Policy and Management, Public Health and Nursing, Universitas Gadjah Mada, Yogyakarta, Indonesia, **8** Research Institute for Health Sciences, Chiang Mai University, Chiang Mai, Thailand

* susanna.hutajulu@ugm.ac.id

## Abstract

### Background

Breast cancer is a significant public health concern worldwide, including in Indonesia. Little is known about the spatial and temporal patterns of breast cancer incidence in Indonesia. This study aimed to analyze temporal and spatial variations of breast cancer incidence in Yogyakarta Province, Indonesia.

### Methods

The study used breast cancer case data from the Yogyakarta Population-Based Cancer Registry (PBCR) from 2008 to 2019. The catchment areas of the PBCR included the 48 sub-districts of 3 districts (Sleman, Yogyakarta City, and Bantul). Age-standardized incidence rates (ASR) were calculated for each subdistrict. Joinpoint regression was used to detect any significant changes in trends over time. Global Moran's and Local Indicators of Spatial Association (LISA) analyses were performed to identify any spatial clusters or outliers.

### Results

The subdistricts had a median ASR of 41.9, with a range of 15.3–70.4. The majority of cases were diagnosed at a late stage, with Yogyakarta City having the highest proportion of diagnoses at stage 4. The study observed a significant increasing trend in breast cancer incidence over the study period the fastest of which is in Yogyakarta City with an average

**Data Availability Statement:** Data cannot be shared publicly because of restrictions imposed by the ethics committee as most of these contain

patient data, albeit de-identified, and it may be possible to determine the identity of participants. Should there be a request for data, this can be sent to the corresponding author (email: susanna. hutajulu@ugm.ac.id) and the institutional ethics committee (email: mhrec_fmugm@ugm.ac.id) at Universitas Gadjah Mada, Indonesia, with data access queries as well.

**Funding:** The activities of Yogyakarta population-based cancer registry and Dr Sardjito hospital-based cancer registry were supported by the annual budget of Dr Sardjito General Hospital, the Faculty of Medicine, Public Health and Nursing, Universitas Gadjah Mada, and the Provincial Health Office, Yogyakarta. The strengthening of Yogyakarta population-based cancer registry was also supported by the European Society for Medical Oncology (ESMO) in collaboration with the International Agency for Research on Cancer (IARC) through the Evaluating Medical Oncology Outcome (EMOO) in Asia Study (year 2019-2021). The funders had no role in study design, data collection and analysis, decision to publish, or preparation of the manuscript.

**Competing interests:** The authors have declared that no competing interests exist.

annual percentage change of 18.77%, with Sleman having an 18.21% and Bantul having 8.94% average changes each year (p <0.05). We also found a significant positive spatial autocorrelation of breast cancer incidence rates in the province (I = 0.581, p <0.001). LISA analysis identified 11 subdistricts which were high-high clusters in the central area of Yogyakarta City and six low-low clusters in the southeast region of the catchment area in the Bantul and Sleman Districts. No spatial outliers were identified.

## Conclusions

We found significant spatial clustering of BC ASR in the Yogyakarta Province, and there was a trend of increasing ASR across the region. These findings can inform resource allocation for public health efforts to high-risk areas and develop targeted prevention and early detection strategies. Further res is needed to understand the factors driving the observed temporal and spatial patterns of breast cancer incidence in Yogyakarta Province, Indonesia.

## Introduction

Breast cancer (BC) is the most common type of cancer among women in Indonesia, comprising 30.8% of all female cancer cases in 2020 and causing 20.4% of female cancer-related deaths in the same year. Lung cancer is the only cancer type with a higher mortality. In 2040, it is predicted that there will be a 47.1% and 62.1% increase in the incidence and mortality, respectively, of female BC in Indonesia [1]. Based on the national health survey, the Yogyakarta Province, one of the 38 provinces of Indonesia, has more than twice the national cancer prevalence [2], with BC incidence ranked first in this region based on the population-based cancer registry data (PBCR) [3].

Cancer epidemiology studies have extensively used trend analysis, which can provide valuable information regarding cancer disease magnitude through time. Joinpoint regression can characterize trends using joined linear segments on a logarithmic scale, and the point where two segments meet is called a "joinpoint." This approach has been extensively applied to assess long-term trends in cancer incidence to call for health system efficiency and effectiveness evaluation, policy reform, and further improved cancer control and prevention initiatives [4].

Numerous individual risk factors have been linked to BC occurrence, including diet, lifestyle, body mass index (BMI), reproductive history, comorbidity, and genetic factors [5–7]. Besides individual risk factors, many studies have explored and found geographical differences in BC incidence, which are linked to urbanization and lifestyle changes [8], regional socioeconomic activities and welfare [9], and environmental exposure [10]. Identifying areas with high BC burden may provide valuable insight concerning preventive and management strategies and help effective resource distribution to where they are most needed [11].

Previous studies around the globe have investigated and found BC incidence disparities based on its geographical distribution [12–17]. However, little is understood regarding this topic in Indonesia. An earlier study had been conducted in Yogyakarta Province [18]. It described the geographical variations of BC incidence among various types of cancers at the district level in 2019–2020 using data from four hospital-based registries. The use of PBCR data, exploration at the subdistrict level, and analysis of temporal dynamics and geographical heterogeneity in a long period of observation can picture a more realistic disease burden. The present study aimed to determine the dynamic patterns and geographical variations of BC at the subdistrict levels using PBCR data on cases diagnosed from 2008 to 2019. The study findings will provide insights into the epidemiology of BC in the region and stimulate the development of more effective targeted public health strategies.

## Methods

### Study design and population

We performed a cross-sectional study to examine the incidence of BC, defined as C50.0–9 according to International Classification of Diseases and Related Health Problems– 10th Revision (ICD-10) codes, in the Yogyakarta Province. This province comprises five districts, namely Kulon Progo District, Sleman District, Yogyakarta City, the capital of Yogyakarta Province, Bantul District, and Gunungkidul District, spanning 3,186 km$^2$, with a population of 3.7 million in 2021. We collected data from the Yogyakarta PBCR which covers three catchment districts, namely Sleman, Yogyakarta City, and Bantul, out of the five districts of the Yogyakarta Province as determined by the Indonesian Ministry of Health. Sleman District covers an area of 574.8 km$^2$, Yogyakarta City covers an area of 32.5 km$^2$, and Bantul District covers an area of 506.8 km$^2$. These areas have a total of 48 subdistricts, the smallest integrated administrative unit in which case and population statistics were collected. The name and locations of subdistricts within these catchment regions are provided in Fig 1.

In 2016, the Indonesian Ministry of Health established a national PBCR network comprising 14 out of the then 34 provinces of Indonesia (4 added as of 2022). The Yogyakarta PBCR is part of the network, which has gathered data of medical records from 18,992 cancer cases from several primary healthcare facilities, pathology laboratories, referral hospitals and cancer clinics diagnosed between January 2008 and December 2020. The Yogyakarta PBCR network operates under a national decree from The Indonesian Ministry of Health and further provincial clearance. In each health facility, each patient has given written general informed consent upon admission, which included that the patient's data may be utilized for research purposes. In the present study, the data obtained from the PBCR have been fully anonymized for each patient. The joint ethics committee from the Faculty of Medicine, Public Health and Nursing, Universitas Gadjah Mada/Dr Sardjito General Hospital, Yogyakarta has acknowledged and agreed on this use of secondary data (ethical clearance reference number: KE/FK/1362/EC/2022).

The variables registered in the PBCR database are based on the adapted version of CanReg5 for the Indonesian cancer registry (version 5.00.40). The first part of the database included sociodemographic variables such as name, sex, ethnicity, date of birth, citizen card number, permanent resident address, occupation, marital status, and religion. The second part of the database included clinical and pathological data such as age at diagnosis, the basis of diagnosis, tumor morphology, topography and behavior based on the International Classification of Diseases for Oncology (ICD-O), tumor extent (SEER Summary Staging Manual), stage and metastasis status, and treatment. The last part of the database included was data summary and follow-up results from physician records, vital signs and examination results, date of the first identification and last date of contact. Case recording was performed in adherence to the Manual for Cancer Registry Personnel from the IARC-WHO.

The present study included all data of female patients with BC aged 20 years and older diagnosed during 2008–2019 who resided in the district for at least six months. Data extraction from the PBCR database was done between May and July 2022. During the twelve years observed, 4,268 cases of BC were recorded. The patient variables included are permanent address, year of diagnosis, sex, age distribution, the basis of diagnosis, and stage.

### Data sources, variables, and measurements

We obtained population data for Sleman, Yogyakarta City, and Bantul Districts at the subdistrict level from the Central Bureau of Statistics of Yogyakarta Province. We calculated the age-

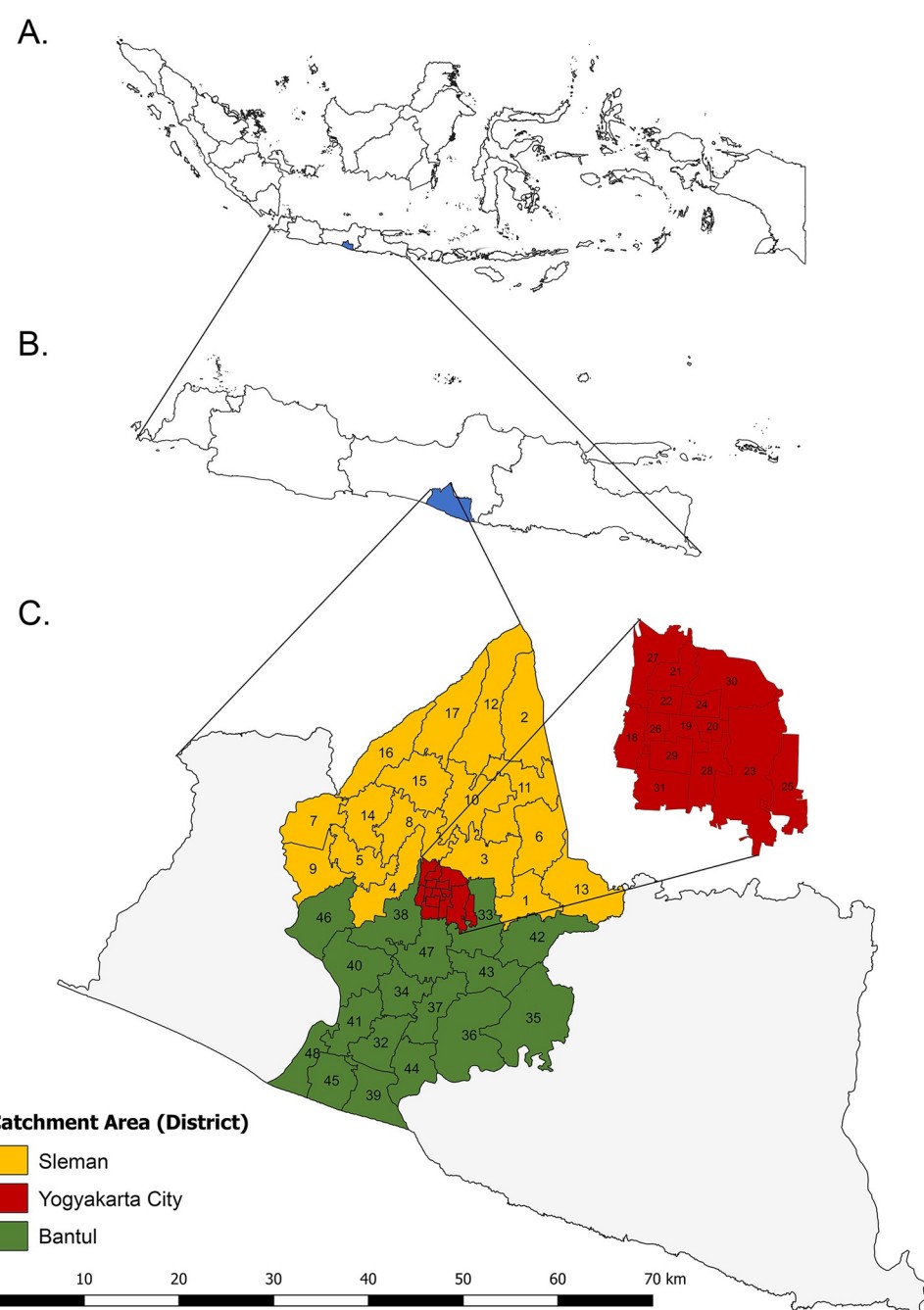

**Fig 1. District and subdistrict regions of the Yogyakarta Province, and details of study catchment area.** (A) Map of Indonesia country. Yogyakarta Province is indicated in the blue area; (B) Map of Java Island. Yogyakarta Province is indicated in the blue area; (C) Map of Yogyakarta Province and study catchment area (Sleman District, Yogyakarta City, and Bantul District). List of Sleman District's subdistrict: 1) Berbah, 2) Cangkringan, 3) Depok, 4) Gamping, 5) Godean, 6) Kalasan, 7) Minggir, 8) Mlati, 9) Moyudan, 10) Ngaglik, 11) Ngemplak, 12) Pakem, 13) Prambanan, 14) Seyegan, 15) Sleman, 16) Tempel, 17) Turi. List of Yogyakarta City's subdistrict: 18) Wirobrajan, 19) Gondomanan, 20) Pakualaman, 21) Jetis Kota, 22) Gedongtengen, 23) Umbulharjo, 24) Danurejan, 25) Kotagede, 26) Ngampilan, 27) Tegalrejo, 28) Mergangsan, 29) Kraton, 30) Gondokusuman, 31) Mantrijeron. List of Bantul District's subdistrict: 32) Bambanglipuro, 33) Banguntapan, 34) Bantul, 35) Dlingo, 36) Imogiri, 37) Jetis, 38) Kasihan, 39) Kretek, 40) Pajangan, 41) Pandak, 42) Piyungan, 43) Pleret, 44) Pundong, 45) Sanden, 46) Sedayu, 47) Sewon, 48) Srandakan. Shapefile was obtained and adapted from Indonesia's Geospatial Information Agency (Badan Informasi Geospasial) (https://tanahair.indonesia.go.id/).

specific risk of BC in the population in 5-year age increments (up to 79 and $\geq$ 80 years of age) using these population data. We then calculated the age-standardized incidence rates (ASR) with the World Standard Population [19] as weights reference and reported the ASR per 100,000 person-year. The ASRs were determined using population data from 2014, the central year of the study.

## Statistical analysis

Joinpoint regression was performed to determine the dynamic variations in ASRs across the observation years. It represents the time series using a few continuous linear segments linked at points indicating the year that a statistically significant shift in the rate's trend occurred [20]. Average annual percent of change (AAPC), a summary of the average changes of ASR, is computed based on the Jointpoint model for the observed timeline. AAPC will be described over the total observation years and the period with an identified Joinpoint. The Sleman, Yogyakarta City, and Bantul base map shapefile were obtained from Indonesia's Geospatial Information Agency (Badan Informasi Geospasial) and made available to the public for use, adaptation, and distribution at a website (https://tanahair.indonesia.go.id/). A breast cancer ASR pattern map, in which the subdistricts were classified into quintiles based on their BC ASR and colored according to their quintile, was created, with the subdistricts classed as the highest (fifth) quintiles being identified as hotspots.

We used Moran I's statistics, one of the most widely known and used methods for investigating spatial autocorrelation in health studies [21]. The Global Moran's I statistic was used to determine if there is global clustering or spatial autocorrelation in the pattern of BC ASR. Values of Moran's index (I) range from −1 to +1. The further away the value from zero, the stronger the spatial autocorrelation. When the value of I is greater than zero, the distribution has a positive spatial autocorrelation, which means that the value in a spatial unit, which is at the subdistrict level, tends to be similar to those in adjacent subdistricts. On the contrary, when the value of I is less than zero, the distribution has a negative spatial autocorrelation, which means that the value in a subdistrict tends to be dissimilar to those in subdistricts [22]. Compared to similar analysis, this result is relatively simple and easy to interpret.

A limitation of the Global Moran's I statistic is that it may not fully capture the extent of spatial clustering or dispersion in a dataset because it represents a summary measure of the entire study area, and it does not reveal the presence of localized clusters or outliers. To identify local statistically significant clusters and outliers, analysis with Local Indicators of Spatial Association (LISA) was used. A Local Moran's I statistic is calculated for each spatial unit, which is at the subdistrict level, allowing categorization of each subdistrict. Four types of spatial clusters or outliers may be identified using the LISA statistic: (1) High-high cluster (subdistrict with high ASR located adjacent to subdistricts with also high ASR); (2) Low-low cluster (subdistrict with low ASR located adjacent to subdistricts with also low ASR); (3) High-low outlier (subdistrict with high ASR located adjacent to subdistricts with low ASR); and (4) Low-high outlier (subdistrict with low ASR located adjacent to subdistricts with high ASR). High-high and low-low clusters make spatial autocorrelation more positive. In contrast, high-low and low-high outliers make spatial autocorrelation more negative [23]. A choropleth map was generated coloring the subdistricts which were statistically significant according to the LISA analysis.

Joinpoint analysis was performed using Joinpoint Regression Program version 4.9.1.0 (SEER, USA). The BC ASR quintile maps were generated using a GIS open-source software, the Quantum Geographic Information System program (QGIS) Desktop version 3.26. Statistical analyses, including spatial statistics tests, were performed using R statistical software

version 4.2.2. Global Moran analysis was conducted using R software spdep package [24], whereas LISA analysis was performed using the R software rgeoda package [25].

## Results

### Data quality and BC ASR

To check for PBCR data quality, the percentage of morphology verification (%MV) was used, which was calculated to be 73.81% overall (S1 Table). The ASR of BC in the three districts during 2008–2019 was 41.35 per 100,000 person-year. The incidence was found to be the highest in Yogyakarta City, followed by Sleman and Bantul Districts (ASR = 54.51, 41.89, and 34.75, respectively).

### Age distribution and trend of age at diagnosis

The median age of patients with BC in Yogyakarta Province was 51 years old, with the median of each district ranging from 50 to 53 years old. Less than 10% of the cases were diagnosed younger than 30 or older than 70 years old. The peak incidence of BC was in the 60–64 age group (ASR = 80.00) as shown in Fig 2. Peak incidence was in the 50–64 age group in Sleman (ASR = 66.59), the 60–64 age group in Yogyakarta City (ASR = 120.81), and the 50–54 age group in Bantul (ASR = 66.59). The peak incidence shifted from the 50–54 age group in 2008–2013 to the 60–64 age group in 2014–2019 both in Sleman and Yogyakarta City. The shift was also identified from the 55–59 age group to the 50–54 and 60–64 age groups in Bantul (Fig 3).

### Stage distribution of BC cases

Information on the stage at diagnosis was limited, with the percentage of missing data ranging from 73.55%-77.61% in each district. As presented in Fig 4, most of the cases were diagnosed at a later stage. Among cases with a recorded stage, 53.28% were diagnosed at stage 4 in

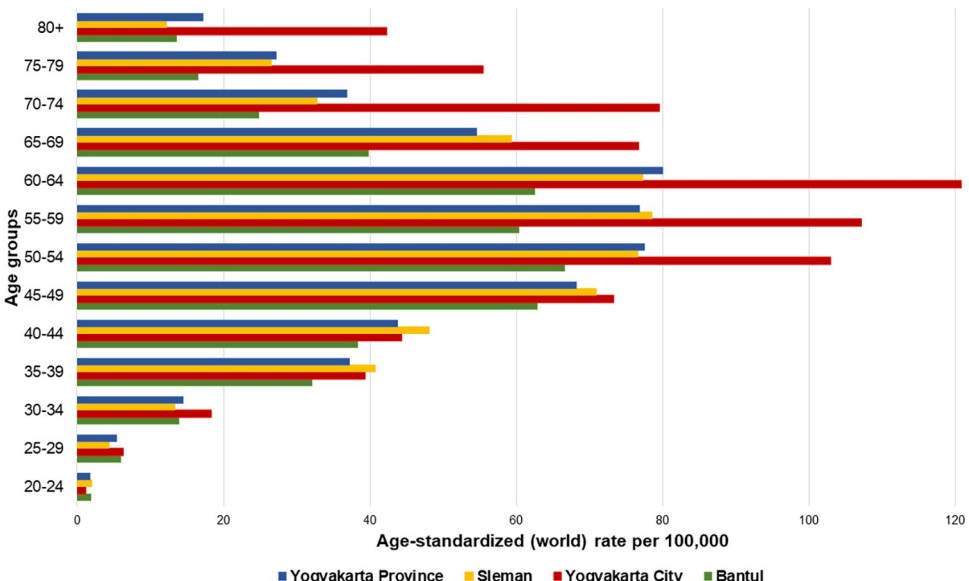

**Fig 2. Breast cancer ASR based on age groups in the three catchment districts of Yogyakarta Province during 2008–2019.** The peak incidence of BC in the province was at 60–64 age group (ASR = 80.00). Peak incidence in the individual district is as follow: Sleman at 50–64 age group (ASR = 66.59), Yogyakarta City at 60–64 age group in (ASR = 120.81), and Bantul at 50–54 age group (ASR = 66.59).

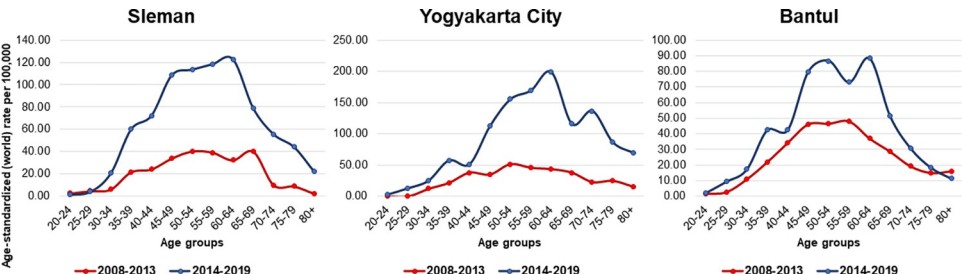

**Fig 3. Age trend of breast cancer cases in each district in the Yogyakarta Province during 2008–2019.** The age trend showed that peak incidence shifted from 50–54 in 2008–2013 to 60–64 age group in 2014–2019 both in Sleman and Yogyakarta City. In Bantul the shift was observed from 55–59 to 50–54 and 60–64 age groups.

Yogyakarta City. Although the proportion of cases in stage 4 was lower in Sleman (40.85%) and Bantul (47.32%), it remained the biggest proportion in each district.

## Temporal trend of BC in Yogyakarta during 2008–2019

As shown in Fig 5, BC incidence in Yogyakarta Province increased steadily throughout the twelve years (AAPC = 14.7%; 95% CI = 8.9–20.9%; $p < 0.05$). The increasing trend of BC was apparent in all districts (AAPC of Sleman = 18.21%, Yogyakarta City = 18.77%, and Bantul = 8.94%; $p < 0.05$). In Sleman and Yogyakarta City, we discovered a significant increase in BC incidence during 2012–2017 (AAPC = 36.09 and 34.05%, respectively).

## Geospatial distribution of BC incidence

The highest BC ASR was observed in the Pakualaman Subdistrict (subdistrict number 20; ASR = 70.38) of the Yogyakarta City and the lowest was in the Dlingo Subdistrict (35;

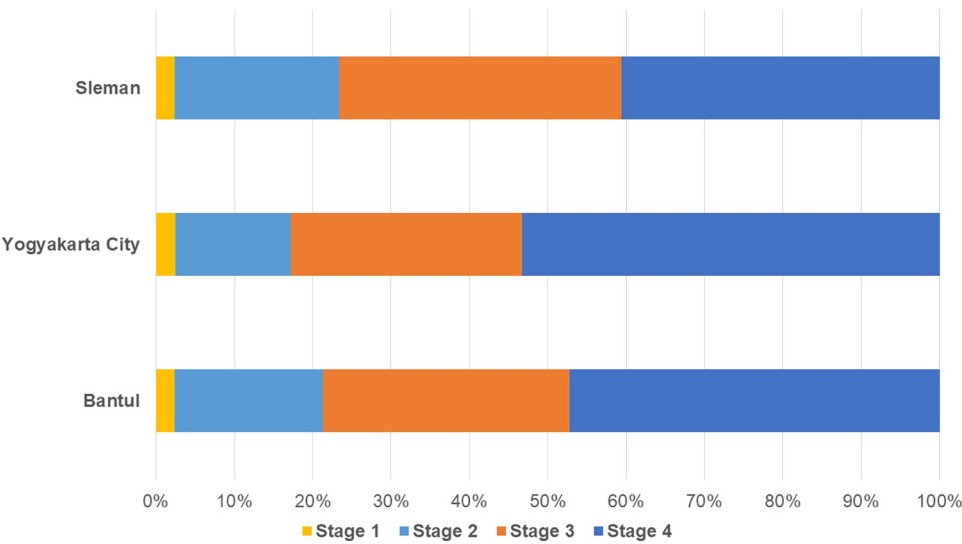

**Fig 4. Distribution of the stage at diagnosis of breast cancer cases in each district in the Yogyakarta Province during 2008–2019.** Stage at diagnosis was provided among cases with known data as follows: Sleman, stage 1 = 2.37%, stage 2 = 18.97%, stage 3 = 31.42%, stage 4 = 47.23%; Yogyakarta City, stage 1 = 2.46%, stage 2 = 14.75%, stage 3 = 29.51%, stage 4 = 53.28%; Bantul, stage 1 = 2.42%, stage 2 = 20.91%, stage 3 = 36.06%, stage 4 = 40.61%. Vast majority of cases in the three districts were diagnosed at an advanced stage. Yogyakarta City has the highest of patients with stage 4 disease, followed by Bantul and Sleman.

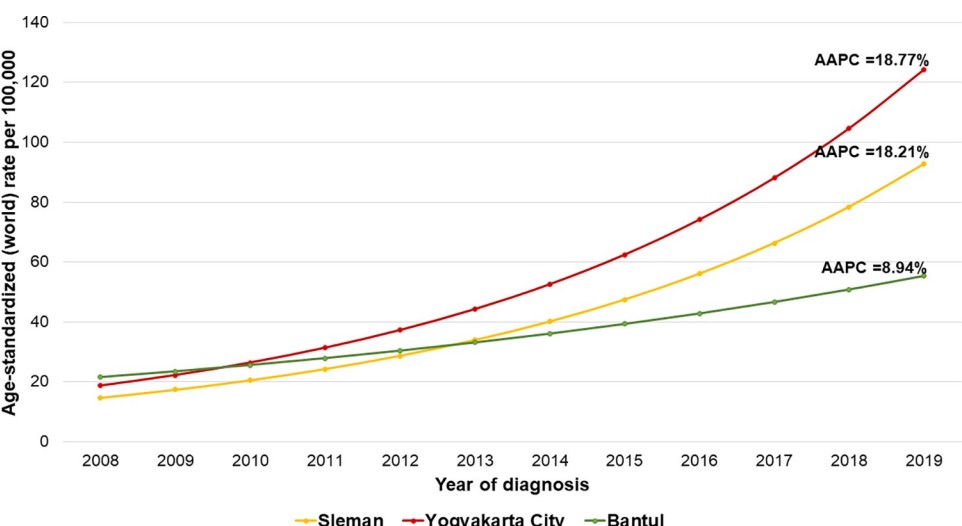

**Fig 5. Average annual percent of change (AAPC) of breast cancer incidence in the three districts during 2008–2019.** All three districts showed an increasing trend of breast cancer incidence with average annual percentage change (AAPC) in Sleman 18.21%, Yogyakarta City 18.77%, and Bantul 8.94% (p<0.05). A significant incline of breast cancer incidence was clearly seen in Sleman and Yogyakarta City during 2012–2017 (AAPC = 36.09 and 34.05%).

ASR = 15.35) of the Bantul District (S2 Table). The geographic variation of BC ASR for 2008–2019 is shown in Fig 6A. The hotspot for BC ASR quintiles were found to be 8 of 14 subdistricts of Yogyakarta City, namely Pakualaman (subdistrict number 20; ASR = 70.38), Kraton (29; ASR = 68.34), Danurejan (24; ASR = 65.99), Mantrijeron (31; ASR = 62.45), Umbulharjo (23; ASR = 61.38), Gondomanan (19, ASR = 59.62), Mergangsan (28; ASR = 58.91), and Ngampilan Subdistricts (26; ASR = 54.03), in Depok Subdistrict (3; ASR = 62.70) of Sleman District, and in Bantul Subdistrict (34; ASR = 52.88) of Bantul District. Hotspots were generally located in the central region of the study catchment area.

## Global and Local Spatial Autocorrelation (LISA)

The global spatial autocorrelation analysis of the cumulative ASR of incidence in Sleman, Yogyakarta City, and Bantul Districts between 2008 and 2019 showed a Moran's I index of 0.581 (z = 6.755; p <0.001), a positive spatial autocorrelation which indicates that the subdistricts tend to have similar BC incidence with its neighbors (i.e. subdistricts with a high BC incidence tend to have neighboring subdistricts with also high incidence rates, while subdistricts with a low BC incidence rate were more likely to have neighboring subdistricts with also low incidence rates).

The LISA analysis of the cumulative ASR of incidence in Sleman, Yogyakarta City, and Bantul Districts between 2008 and 2019 revealed the presence of 11 high-high clusters of BC incidence which included Danurejan (subdistrict number 24), Gedongtengen (22), Gondokusuman (30), Gondomanan (19), Jetis Kota (21), Kraton (29), Mantrijeron (31), Mergangsan (28), Ngampilan (26), Pakualaman (20) and Umbulharjo (23) Subdistricts, all of which are in the Yogyakarta City. These findings mean that these subdistricts have a high ASR and are located adjacent to subdistricts with also high ASR. LISA analysis visualization also demonstrated the presence of six low-low clusters of BC incidence in the southeastern area of the catchment region, including Piyungan (42), Dlingo (35), Pleret (43), Imogiri (36), and Pundong (44) Subdistricts from Bantul District, and Prambanan Subdistrict (13) from the Sleman District. These findings mean these subdistricts have low ASR and are located adjacent to subdistricts with also low ASR. No spatial outliers were found in the analysis (Fig 6B).

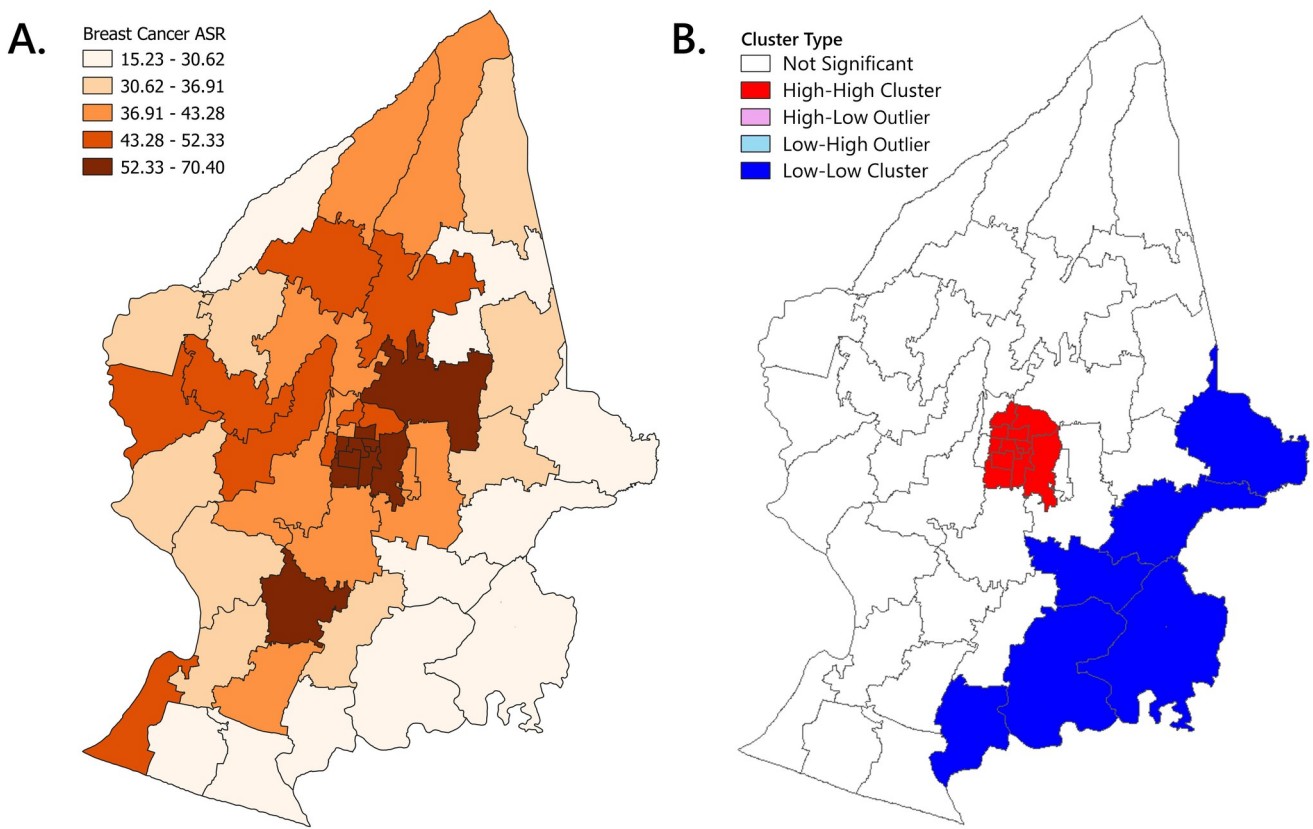

**Fig 6. Breast cancer (BC) incidence map in the catchment area of Sleman, Yogyakarta City and Bantul Districts.** (A) BC age-standardized rate (ASR) quintile map identifying hotspots (fifth quintile) mostly in the central region of the catchment area; and (B) Local indicators of spatial association (LISA) analysis identifying high-high clusters (red area) in the Yogyakarta City and low-low clusters (blue area) in the southeastern area of Sleman and Bantul Districts. Shapefile was obtained and adapted from Indonesia's Geospatial Information Agency (Badan Informasi Geospasial) (https://tanahair.indonesia.go.id/).

## Discussion

This is the first Indonesian study analyzing BC incidence using PBCR data with an extensive observation period. This investigation enhanced the understanding of BC distribution and variation with time and revealed geographic disparities in the Yogyakarta Province. One of the reasons why this study is important is because of the fact that the province has the highest cancer prevalence in the country according to the national survey [26] The study findings may also help identify likely determinants of BC in the region and are necessary for planning and assessing BC control efforts.

Our study's highest BC incidence is in the 50–64 age group, which is similar to that in Japan (45–49 and 60–64 age groups) [27]. The result of our study is higher than in other Asian countries such as South Korea (40–49 age group) [28] and China (50–54 age group) [29] but lower than the Western countries such as the United States (US) (70 years), the United Kingdom (70 years) [30] and Australia (65–69 age group) [27]. The discovery of a general lower age of BC onset in Asian populations has created a debate about whether BC in Asia and the West have different characteristics. Prior studies comparing European and Asian cancer registries demonstrated that this discrepancy was attributable to a substantial cohort effect, resulting in a difference in the age at which BC was first diagnosed [31]. Another study found that estimates of BC onset ages for the most recent cohorts in certain Asian countries were later than those in

the US [32], with other studies attributing this change to a shift toward a more westernized lifestyle, the implementation of BC screening, and increase of lifespan [33,34]. A more recent cohort of our study population may explain why the peak age of BC onset is later than in other Asian studies such as China and South Korea. This has important implications since due to decreased screening, diagnostic, and therapeutic activities, elderly patients with BC typically have poorer survival outcomes [35]. Older patients with BC also have a decline in social functioning [36] and productivity loss [37]. The detection of later peak age of BC onset in our population demands to be further addressed to increase survival and quality of life.

We observed an increased trend in BC incidence (AAPC = 14.7%). This rate is much higher than that of other countries such as Thailand (2.0–4.5%) [38,39], Iran (4.4%) [40], and the already decreasing trend in the US (-0.4%) [41]. Our study also found that the BC ASR in our study catchment area is 41.35 per 100,000 person-year, similar to the GLOBOCAN estimation for Indonesia in 2020 (44.0) [1]. Although higher than the overall BC incidence in Asia (36.8) [42], this number is lower than previously reported data from other countries such as Japan (76.3), South Korea (64.2), Singapore (77.9), South Africa (52.6), Australia (96.0), US (90.3), and the overall ASR from high Human Development Index (HDI) countries (42.7) [30]. One of the major elements contributing to the high incidence of BC in developed nations has been the widespread use of mammography screening [43]. On the contrary, Asian countries including Indonesia have a lower incidence of BC, though it is growing rapidly due to socioeconomic progress and lifestyle changes [34].

There was a clear inequality in the ASR of BC in the three catchment districts of Yogyakarta Province. In particular, subdistricts of Yogyakarta City generally had a higher ASR than Bantul and Sleman. This finding contrasts the earlier study [18] showing a higher BC incidence rate in Sleman than in Yogyakarta City. Reasons for this discrepancy may be the usage of PBCR data and ASRs for calculating incidence in our study.

The Yogyakarta City showed a faster increasing trend of BC incidence compared to other districts (AAPC = 18.77%), with a significant increase observed from 2012 to 2017 (AAPC = 36.09%). We also found that the distribution of BC incidence has a positive spatial autocorrelation (i.e., spatially clustered), with eleven subdistricts classified as high-high clusters located in the central area of study, all of which are subdistricts of Yogyakarta City. These subdistricts included Danurejan (subdistrict number 24), Gedongtengen (22), Gondokusuman (30), Gondomanan (19), Jetis (21), Kraton (29), Mantrijeron (31), Mergangsan (28), Ngampilan (26), Pakualaman (20) and Umbulharjo (23) Subdistricts. Six subdistricts were classified as low-low clusters southeast of the catchment area composed of subdistricts of Sleman and Bantul Districts, including Piyungan (subdistrict number 42), Dlingo (35), Pleret (43), Imogiri (36), Pundong (44), and Prambanan (13) Subdistrict. The Global Moran's I index found in this study (I = 0.581) is higher than the results found in other studies investigating BC incidence, such as those done in Hangzhou, China (I = 0.563) [14], Shenzhen, China (I = 0.372) [17], and in Tennessee, US (I = 0.203) [16]. This pattern suggests that BC incidence in the location of our study has a stronger positive autocorrelation, and therefore appears more clustered. Furthermore, the absence of any spatial outliers shows that subdistricts with large differences in BC ASR were not neighbors with each other, reinforcing the idea of spatial clustering.

Hereditary cancer syndromes are inherited disorders passed throughout generations, with a higher-than-normal risk of developing certain cancer. Examples of hereditary cancer syndromes are BRCA mutations associated with breast cancer. Women with BRCA mutation (BRCA1 or BRCA2) have a substantially 69–72% increased risk of developing breast cancer [44]. BRCA mutation screening has shifted from primarily conducted in breast cancer patients to unaffected women for prevention [45], enabling those women to have intensified screening and prophylactic treatments to prevent and detect breast cancer earlier. Although BRCA

screening has yet to be implemented in our unaffected population, our findings of regions with high BC ASR may serve as a foundation for conducting genetic screening in these high BC ASR areas. Other than possible genetic factors, several local sociodemographic, economic, and environmental characteristics of the three districts may help explain this phenomenon.

The urbanization level of Yogyakarta City has been studied to be higher than Bantul and Sleman [46]. Yogyakarta City consistently had a higher HDI than Bantul and Sleman [47]. More developed populations have been shown to have a higher rate of BC [48–50], which may be due to differences in lifestyle [51].

It has also been established in previous studies that there are significant correlations between BC incidence and higher income levels [52] as well as higher education levels [53]. The Per Capita Expenditure (PCE) and the average education length of people in Yogyakarta City were higher than in Bantul and Sleman [47]. A previous study showed that, in Indonesian women, higher household expenditure and higher education were also positively correlated with being aware of mammography [54].

Another important factor is reproductive behavior. Increased full-term pregnancies are associated with a lower risk of BC [55], whereas nulliparity increases the risk [56]. In a report, women in Yogyakarta City had a higher rate of nulliparity (7.73%) than Sleman (5.26%) and Bantul (5.73%). Women in Yogyakarta City also had a lower number of mean live births than women in Sleman and Bantul [57]. Another report has shown that Yogyakarta City has the lowest rate of infant breastfeeding compared to other districts in the province [58]. Breastfeeding had been shown to be associated with lower BC risk [59]. One local study in Umbulharjo, a Yogyakarta City subdistrict which is a high-high cluster, found that being a working mother is associated with a lower rate of infant breastfeeding. High economic pressure in the urban areas of Yogyakarta City may be one of the reasons for a mother to work and not breastfeed [60].

Smoking has been observed as a risk factor for BC [61]. Although the prevalence of smokers in women in Yogyakarta Province is not known, a survey showed that among women who do smoke in the province, cigarette consumption per week was higher in Yogyakarta City (53.2) than in Bantul (17.5) and Sleman (23.1) [62]. In addition, never-smoker women are at risk of secondhand smoke, which has been found to increase the risk of BC [63]. A study found that 1 in 14 cases of BC could have been prevented by eliminating secondhand smoking [64]. The fact that almost 24% of the population aged >15 years of Yogyakarta Province are smokers [65] may imply a high proportion of secondhand smokers among women in the region.

Air pollution is a potential risk factor for BC [66]. Local investigation in Yogyakarta Province observed that, in comparison to other districts, subdistricts in the Yogyakarta City have the greatest concentration of particulate matter PM10 in the air [67], which is linked with increased BC incidence [68]. The high amount of economic operations and public facilities demanding transportation in Yogyakarta City is likely accountable for the significant PM10 production.

Although a BC screening program has been implemented in Indonesia since 2007 [69], this program was performed in a symptom-based manner. Upon clinical breast examination (CBE), patients with abnormal findings will be referred and provided with radiographic examination based on the Minister of Health regulations [70]. CBE is linked with 17% to 47% downstaging of BC [71]. Women's willingness to undergo CBE is influenced by multiple factors, such as marital status, perceived susceptibility, perceived barriers, and self-care [72]. This might explain the fact that only 9.8% of the targeted Indonesian women had been screened by 2018 [69]. This low coverage may also reflect the situation in our province. Another local study found that women in Yogyakarta Province have higher perceived barriers to BC screening than other provincial regions [73].

## Study strength and limitation

The strength of the current study derives from the utilization of 12 years' data of BC incidence from the provincial PBCR. To guarantee data coverage and quality, case discovery and data collecting followed a structured method which complied with the requirements for Cancer Incidence in 5 Continents (CI5). The Yogyakarta PBCR data we utilized in this analysis contained a small number of unknown ages at diagnosis (0.03%), ill-defined sites (0.78%), and unknown primary sites (2.06%). On the other hand, a limitation of our study is that the PBCR data in the Bantul District and Yogyakarta City had a 67.35% and a 73.58% morphological verification rate, which are below the level advised by the World Health Organization (>75%). This is in contrast to Sleman, which had a morphological verification rate of 78.16%. Another drawback is that the PBCR does not cover all patients with BC in Yogyakarta Province because the PBCR only covers 3 out of 5 districts and some BC occurrences may have been overlooked. However, this should not impact the BC cluster detection and LISA analyses.

## Conclusions

Our findings demonstrate an increased BC incidence throughout 2008–2019 in all subdistricts observed. This work also found that BC incidence distribution is uneven and spatially clustered in the Yogyakarta City. Differences in urbanization, income and education level, reproductive behavior, cigarette consumption, and air pollution might contribute to the ASR differences and case clustering. Understanding the causes of increased incidence and clustering in specific subdistricts relative to others requires more in-depth investigation. Our findings may inform resource allocation for public health efforts to control BC in Yogyakarta Province. To implement BC screening programs and further promote healthcare services to offer better management for patients with BC, a countrywide study may provide better and more thorough data.

## Supporting information

**S1 Table. Age-standardized rate of breast cancer among subdistricts level in the three catchment districts of Yogyakarta Province (2008–2019).**
(PDF)

**S2 Table. Breast cancer data quality in Yogyakarta PBCR through data collection period (2016–2022).**
(PDF)

**S1 Dataset. Minimal dataset.**
(XLSX)

## Acknowledgments

We gratefully thank Erik Hookom for language editing.

## Author Contributions

**Conceptualization:** Bryant Ng, Herindita Puspitaningtyas, Juan Adrian Wiranata, Patumrat Sripan.

**Data curation:** Bryant Ng, Herindita Puspitaningtyas, Juan Adrian Wiranata, Susanna Hilda Hutajulu.

**Formal analysis:** Bryant Ng, Herindita Puspitaningtyas, Juan Adrian Wiranata, Susanna Hilda Hutajulu, Guardian Yoki Sanjaya, Lutfan Lazuardi, Patumrat Sripan.

**Funding acquisition:** Susanna Hilda Hutajulu.

**Investigation:** Bryant Ng, Herindita Puspitaningtyas, Juan Adrian Wiranata, Susanna Hilda Hutajulu.

**Methodology:** Bryant Ng, Herindita Puspitaningtyas, Juan Adrian Wiranata, Susanna Hilda Hutajulu, Guardian Yoki Sanjaya, Lutfan Lazuardi, Patumrat Sripan.

**Project administration:** Herindita Puspitaningtyas.

**Resources:** Susanna Hilda Hutajulu, Irianiwati Widodo, Nungki Anggorowati, Guardian Yoki Sanjaya, Lutfan Lazuardi.

**Software:** Bryant Ng, Herindita Puspitaningtyas, Juan Adrian Wiranata.

**Supervision:** Susanna Hilda Hutajulu, Irianiwati Widodo, Nungki Anggorowati, Guardian Yoki Sanjaya, Lutfan Lazuardi, Patumrat Sripan.

**Validation:** Bryant Ng, Herindita Puspitaningtyas, Juan Adrian Wiranata, Susanna Hilda Hutajulu, Irianiwati Widodo, Nungki Anggorowati, Guardian Yoki Sanjaya, Lutfan Lazuardi, Patumrat Sripan.

**Visualization:** Bryant Ng, Herindita Puspitaningtyas, Juan Adrian Wiranata.

**Writing – original draft:** Bryant Ng, Herindita Puspitaningtyas, Juan Adrian Wiranata, Susanna Hilda Hutajulu.

**Writing – review & editing:** Bryant Ng, Herindita Puspitaningtyas, Juan Adrian Wiranata, Susanna Hilda Hutajulu, Irianiwati Widodo, Nungki Anggorowati, Guardian Yoki Sanjaya, Lutfan Lazuardi, Patumrat Sripan.

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
