## [Decision Letter · Decision Letter 0]

13 Apr 2023

PONE-D-23-05461Breast cancer incidence in Yogyakarta, Indonesia from 2008-2019: a cross-sectional study using trend analysis and geographical information systemPLOS ONE

Dear Dr. Hutajulu,

Thank you for submitting your manuscript to PLOS ONE. After careful consideration, we feel that it has merit but does not fully meet PLOS ONE’s publication criteria as it currently stands. Therefore, we invite you to submit a revised version of the manuscript that addresses the points raised during the review process.

We look forward to receiving your revised manuscript.

Kind regards,

Abdulkader Murad, Ph.D

Academic Editor

PLOS ONE

Journal Requirements:

3. We note that Figure 1 and 6 in your submission contain [map/satellite] images which may be copyrighted. All PLOS content is published under the Creative Commons Attribution License (CC BY 4.0), which means that the manuscript, images, and Supporting Information files will be freely available online, and any third party is permitted to access, download, copy, distribute, and use these materials in any way, even commercially, with proper attribution. For these reasons, we cannot publish previously copyrighted maps or satellite images created using proprietary data, such as Google software (Google Maps, Street View, and Earth). For more information, see our copyright guidelines: http://journals.plos.org/plosone/s/licenses-and-copyright.

1. You may seek permission from the original copyright holder of Figure 1 and 6 to publish the content specifically under the CC BY 4.0 license.  

"The activities of Yogyakarta population-based cancer registry and Dr Sardjito hospital-based cancer registry were supported by the annual budget of Dr Sardjito General Hospital, the Faculty of Medicine, Public Health and Nursing, Universitas Gadjah Mada, and the Provincial Health Office, Yogyakarta. The strengthening of Yogyakarta population-based cancer registry was also supported by the European Society for Medical Oncology (ESMO) in collaboration with the International Agency for Research on Cancer (IARC) through the Evaluating Medical Oncology Outcome (EMOO) in Asia Study (year 2019-2021). We gratefully thanked Erik Hookom for language editing"

"The activities of Yogyakarta population-based cancer registry and Dr Sardjito hospital-based cancer registry were supported by the annual budget of Dr Sardjito General Hospital, the Faculty of Medicine, Public Health and Nursing, Universitas Gadjah Mada, and the Provincial Health Office, Yogyakarta. The strengthening of Yogyakarta population-based cancer registry was also supported by the European Society for Medical Oncology (ESMO) in collaboration with the International Agency for Research on Cancer (IARC) through the Evaluating Medical Oncology Outcome (EMOO) in Asia Study (year 2019-2021).

Reviewers' comments:

Reviewer's Responses to Questions

**Comments to the Author**

1. Is the manuscript technically sound, and do the data support the conclusions?

Reviewer #1: Partly

Reviewer #2: Yes

2. Has the statistical analysis been performed appropriately and rigorously? 

Reviewer #1: Yes

Reviewer #2: Yes

3. Have the authors made all data underlying the findings in their manuscript fully available?

Reviewer #1: No

Reviewer #2: No

4. Is the manuscript presented in an intelligible fashion and written in standard English?

Reviewer #1: Yes

Reviewer #2: Yes

5. Review Comments to the Author

Reviewer #1: Summary of the research:

The study aims to provide results on temporal and spatial variations of the breast cancer (BC) incidence for 3 districts of Yogyakarta, a province in Indonesia. Primary database are the Yogyakarta Population-Based Cancer Registry (PBCR) and the Central Bureau of Statistics of Yogyakarta province (for population data). Breast cancer is reported as C50.0-9 according to ICD-10, incidence is reported for women with BC aged 20 years and older diagnosed between 2008-2019. Incidence rates (ASR) were age-standardized to the world-standard and are depicted per 100.000 population. Little is known on BC incidence over time and on differences between regions of Indonesia.

An increasing trend of breast cancer incidence is reported, large differences between geographic areas were identified.

As in most countries of the world, breast cancer became a common cancer among women in this province. Reported incidence rates in most recent years are higher than other Asian countries, but remain lower than known from population based-cancer registry analyses in Australia, Western European countries and the US. The high number of women recorded with a diagnosis of BC (in absolute terms) at a relatively young median age of 50-53 years and the large proportion registered at a late stage (if stage was reported), underlines the need to contribute good-quality measures for authorities to take preventive action, invest in cancer control and advance research in the field of cancer epidemiology. Thanks to the authors to raise the important topic of using cancer registry data for better evidence in public health.

Whether the relatively newly established registry of Yogyakarta already yields comparable data that are (close to) complete, timely, and reasonably accurate results – the four main dimensions of population-based cancer data quality (after Bray & Parkin), however needs to be discussed more thoroughly.

Accept with major issues to be addressed.

Comments:

Major issues

1. In 2016, the Indonesian Ministry of Health established a national PBCR network. BC is defined as C50.0-9 according to International Classification of Diseases and Related Health Problems 10th Revision (ICD-10) codes. An increasingly more sophisticated working registry increases the chance to monitor quality issues instead of cancer trends. Cancer in Five Continents is mentioned, but no reference is provided, that the PBCR is considered for certain years as a data source for CI5. Therefore, allow me to raise the following main questions:

a. Has the incidence date during study period (2008-2019) been recorded in a uniform manner (e.g. was the incidence date the earliest date of cancer from the medical documents, or is the date of morphological examination the incident date)?

b. Are rules for recording of multiple primary tumors (e.g. IACR/IARC rules) set up and followed over time? Please provide references to the rules in use.

2. Concerning validity: The percentage of morphological verification (%MV) was calculated to be 73.81% overall. Please provide a comparison to BC-%MV from other registries in the PBCR network and BC-%MV over time of your registry. This will allow the reader to understand better the circumstances in which the registry operates.

Minor issues

3. It is mentioned that screening might be rarely used (<10% uptake) in the province. Is the uptake equally low in the different districts and hasn`t increased significantly over time? Incidental diagnosis due to screening may alter the trend results significantly.

4. On page 11, it is stated that “the Yogyakarta province, …, has more than twice the national cancer prevalence [2], “ Why prevalence? About 200tsd. women living with BC in Indonesia that have been diagnose the past 5 years (https://gco.iarc.fr/today/data/factsheets/populations/360-indonesia-fact-sheets.pdf). Does the prevalence refers to that number? Unfortunately, the resources are in Indonesian only, otherwise I would have checked myself.

“…with BC incidence ranked first in this region [3].” This seems to be a number from the hospital-based cancer registry. Please help the reader to understand the different sources and make sure to be clear about incidence and prevalence.

5. Please state, how was stage collected (UICC/TNM classification?). It is mentioned on page 17, lines 223 ff that “information on the stage at diagnosis was limited, with the percentage of missing data ranging from 73.55%-77.61% in each district”. It might be of value to use essential elements of TNM instead, (e.g. provide proportions of T-stages only)

6. Please check the following links in the references http://labdata.litbang.kemkes.go.id/images/download/laporan/RKD/2018/Laporan_Nasional “An error has occurred. The requested page can't be found.”

Reviewer #2: As the authors write, “identifying areas with high BC burden may provide valuable insight concerning preventive and management strategies and help effective resource distribution to where they are most needed” (lines 78-80). This is the reason why I consider this study a valuable tool for developing future health policies in Indonesia. The authors did a good work. However, I think it needs some changes to improve the article. Please, note that some suggestions are in-depth considerations.

I can suggest the following:

- As far as I know, Indonesian population is ethnically complex and surely strongly structured. The population is described in lines 99-106, the database in 126-133 and the sample under study (variables considered) in lines 136-137. but solely as inhabitants of the districts studied. Although the database comprises ethnicity as a variable, why doesn’t the present study consider a ethnic approach, given the importance of genetics in BC?

- Line 77. The geographical distribution of BC should be put in perspective: association does not amount to causal relation (“linked to”). Please, clarify this when you cite references 6, 7 and 8 in lines 77-78. Do these works refer to causality or a simple statistical association?

- In the Statistical Analysis, do the “spatial units” amount to districts or subdistricts? This should be explicit in the text for a better understanding.

- In lines 188-189, spdep package and rgeoda package should be cited as R packages for those readers unfamiliar with R.

- The claim in line 193 “During the twelve years observed, 4,268 cases of BC were recorded” cannot be considered as a result but a description of the sample. In my opinion, it should be better moved to another section where the data sample is described. However, this is only a suggestion and I’d rather the authors decide.

- In line 221 the authors consider the stage distribution of BC cases but as mentioned in lines 136-137, the variables considered for the present study were permanent address, year of diagnosis, and sex. Why this paragraph on stage distribution? Please, clarify this. In my view it would be better to mention in lines 136-137 ALL variables considered and not only the MAIN variables.

- In figure 1, a small map of Java or Indonesia should help those readers unfamiliar to Asia geography where the area under study is.

- In my opinion, results of I’s Moran and LISA could be provided as supplementary information online.

- Districts cited in lines 267-269 should also be cited according to numbers, in order to facilitate the ubication in figure 1. Also, those in lines 272-273.

- I cannot find the results described in lines 278-281.

- In the discussion from lines 317-319, it would be perhaps of help for the authors the reference Solikhah et al. (2019) Awareness Level about Breast Cancer Risk Factors, Barriers, Attitude and Breast Cancer Screening among Indonesian Women. Asian Pac J Cancer Prev. 20(3):877-884. doi: 10.31557/APJCP.2019.20.3.877. here it is concluded that “Urban women had a poorer level of knowledge of breast cancer risk factors compared to women living in more rural areas” (!!). It seems that the grade of “awareness” can affect the incidence of BC. This is also valid for line 349 (cite 49), where “awareness” is explicitly mentioned.

- Line 330. Please, write the numbers of the six districts (names can also be included). Given that much of the discussion can be seen in figure 1-6, as a rule, include indications such as “see figure 3” or so across the whole discussion.

- This comment is both for “Study strength and limitation” and also for the conclusions. Genetics are just once mentioned in the whole text (line 75). It has been widely reported the importance of genetic polymorphism in BC. Also, genetics is strongly correlated with ethnicity. A reflection on this should be made in order to make the readers aware that the authors consider genetics as a risk factor.

-

Other issues:

- Line 106. It reads Fig 1 but in line 209 Figure 2. ¿Fig or Figure?

- Not everyone knows of Indonesian geography. The authors refer sometimes “Yogyakarta province” and sometimes only “Yogyakarta”. Fig 1 seems to suggest that Yogyakarta is a city and Wikipedia cites Yogyakarta as the “capital city” of the region of Yogyakarta. For the European context, a “province” is a territory where several cities can be located and districts use to be only within the cities. Perhaps it is obvious for the authors, but a clarification across the whole text would help many readers.

- Since I searched in pubmed the terms “Indonesia” and “breast cancer”, I found several references of the years 2020-2023 that I am not sure why the authors do not even mention. Please, check this mainly in the introduction.

6. PLOS authors have the option to publish the peer review history of their article (what does this mean?). If published, this will include your full peer review and any attached files.

Reviewer #1: No

Reviewer #2: No

---

## [Author Response · Author response to Decision Letter 0]

2 Jun 2023

Response to Reviewers

Dear Editor and Reviewer,

We are thankful for the positive feedback received from the editorial team and reviewers 1 and 2, and for the opportunity to respond to the constructive points in our accompanying revised manuscript. Please find below our point-by-point response to the feedback outlining, where relevant, the related changes we have made. We have uploaded revised versions of the manuscript as instructed, including both a clean copy and a track changes version, and additional supplementary files made accordingly.

Response to Editor’s comment

Response:

We have checked these requirements ahead of submitting our revised manuscript and confirmed that our manuscript is in accordance to PLOS ONE’s style requirements.

Response:

Thank you for your kind remarks regarding the participant informed consent. The population-based cancer registry works by obtaining data from multiple sources of health facilities, including primary health care and referral hospital. The registry network operates under a national decree from The Indonesian Ministry of Health and further provincial clearance. In each health facility, each patient has given written general consent upon admission, which included that the patient's data may be utilized for research purposes. In the present study, the data obtained from the PBCR have been fully anonymized for each patient. The joint ethics committee from the Faculty of Medicine, Public Health and Nursing, Universitas Gadjah Mada/Dr Sardjito General Hospital, Yogyakarta has acknowledged and agreed on this use of secondary data (ethical clearance reference number: KE/FK/1362/EC/2022). Furthermore, acknowledging your remarks, we have added this information to the manuscript in lines 129-137 in method section and in the online submission.

3. We note that Figure 1 and 6 in your submission contain [map/satellite] images which may be copyrighted. All PLOS content is published under the Creative Commons Attribution License (CC BY 4.0), which means that the manuscript, images, and Supporting Information files will be freely available online, and any third party is permitted to access, download, copy, distribute, and use these materials in any way, even commercially, with proper attribution. For these reasons, we cannot publish previously copyrighted maps or satellite images created using proprietary data, such as Google software (Google Maps, Street View, and Earth). For more information, see our copyright guidelines: http://journals.plos.org/plosone/s/licenses-and-copyright.

1. You may seek permission from the original copyright holder of Figure 1 and 6 to publish the content specifically under the CC BY 4.0 license. 

Response:

Thank you for highlighting this issue. We have communicated with the Faculty of Geography, Universitas Gadjah Mada, where we obtained the shape file used to create Figures 1 and 6. It is confirmed that the shape file was obtained from the Geospatial Information Agency (Badan Informasi Geospasial/BIG), an official government agency of Indonesia, through their public domain official website: http://tanahair.indonesia.go.id

On this website, BIG has stated in Bahasa Indonesia (Indonesian language) that “users are permitted and free to download, distribute, adapt or make derivatives of basic geospatial information on this website, provided that the information/data source comes from BIG. Users are not allowed to resell any data obtained from this portal”. We provided a screenshot of those statements on this website:

We have complied with the permission to download, modify, and distribute these figures (https://tanahair.indonesia.go.id/portal-web/termcondition). For this clarification, we have revised our manuscript in the:

- Lines 169-171, to update the attribution and citation of the shapefile material

- Figure 1 legend (lines 121-123), to give attribution of the source in the figure legend

- Figure 6 legend (lines 302-304), to give attribution of the source in the figure legend

We thank you for your remarks and attention regarding this matter.

"The activities of Yogyakarta population-based cancer registry and Dr Sardjito hospital-based cancer registry were supported by the annual budget of Dr Sardjito General Hospital, the Faculty of Medicine, Public Health and Nursing, Universitas Gadjah Mada, and the Provincial Health Office, Yogyakarta. The strengthening of Yogyakarta population-based cancer registry was also supported by the European Society for Medical Oncology (ESMO) in collaboration with the International Agency for Research on Cancer (IARC) through the Evaluating Medical Oncology Outcome (EMOO) in Asia Study (year 2019-2021). We gratefully thanked Erik Hookom for language editing"

"The activities of Yogyakarta population-based cancer registry and Dr Sardjito hospital-based cancer registry were supported by the annual budget of Dr Sardjito General Hospital, the Faculty of Medicine, Public Health and Nursing, Universitas Gadjah Mada, and the Provincial Health Office, Yogyakarta. The strengthening of Yogyakarta population-based cancer registry was also supported by the European Society for Medical Oncology (ESMO) in collaboration with the International Agency for Research on Cancer (IARC) through the Evaluating Medical Oncology Outcome (EMOO) in Asia Study (year 2019-2021).

Response:

Thank you for your kind remarks. We have removed the funding information of the study within the manuscript and included amended statements within our cover letter. Thanks also for changing the online submission form on our behalf. We would like to have an updated Funding Statement as read above.

5. The PLOS Data policy requires authors to make all data underlying the findings described in their manuscript fully available without restriction, with rare exception (please refer to the Data Availability Statement in the manuscript PDF file). The data should be provided as part of the manuscript or its supporting information, or deposited to a public repository. For example, in addition to summary statistics, the data points behind means, medians and variance measures should be available. If there are restrictions on publicly sharing data – e.g. participant privacy or use of data from a third party—those must be specified.

Response:

We have currently provided the supporting data (the age-standardized rate at the subdistrict level) behind the joinpoint regression analysis, Moran I's statistics, and LISA results in the supplementary information. Following your remarks, we added supplementary information regarding the data used for Figures 2-4 as minimal dataset attached to the manuscript draft.

Reviewer#1

Major issue

1. In 2016, the Indonesian Ministry of Health established a national PBCR network. BC is defined as C50.0-9 according to International Classification of Diseases and Related Health Problems 10th Revision (ICD-10) codes. An increasingly more sophisticated working registry increases the chance to monitor quality issues instead of cancer trends. Cancer in Five Continents is mentioned, but no reference is provided, that the PBCR is considered for certain years as a data source for CI5. Therefore, allow me to raise the following main questions:

a. Has the incidence date during study period (2008-2019) been recorded in a uniform manner (e.g. was the incidence date the earliest date of cancer from the medical documents, or is the date of morphological examination the incident date)?

b. Are rules for recording of multiple primary tumors (e.g. IACR/IARC rules) set up and followed over time? Please provide references to the rules in use.

Response:

Thank you for pointing out this concern. We confirm that although Yogyakarta, as part of the Indonesia PBCR, has submitted cancer data of 2008-2012 and 2013-2017 periods to the two latest subsequent data calls, the national data has not met the data quality standard to be included in the CI5.

Cancer data in the Yogyakarta PBCR were recorded in accordance to the national consensus, adapting from the Manual for Cancer Registry Personnel (IARC-WHO). Generally, date of diagnosis is recorded based on the earliest available morphological examination result supporting the cancer diagnosis. Date of diagnosis of patients who had histological examination from tissue biopsy following cytological examination will be recorded based on the date of their cytological examination as long as there are no changes in diagnosis. On cases where patient did not have pathological diagnosis, the date of radiological examination will be used as the date of (clinical) diagnosis.

Rules in recording of multiple primary tumors based on the International Rules for Multiple Primary (IARC-WHO) were adhered accordingly. This information has now been added in the manuscript in lines 147-148.

2. Concerning validity: The percentage of morphological verification (%MV) was calculated to be 73.81% overall. Please provide a comparison to BC-%MV from other registries in the PBCR network and BC-%MV over time of your registry. This will allow the reader to understand better the circumstances in which the registry operates.

Response:

Thank you for raising this concern. We acknowledge that comparison of %MV among different registries will be beneficial to elucidate the current circumstance of the registry. However, the status of the other regional registries in the national network are currently not readily accessible, making it difficult to perform the suggested evaluation.

In line with your suggestion, we performed comparison of BC-%MV through-out different period of data collection and observe improvement of BC-%MV in the more recent period. This result is added as supplementary material in S2 Table.

Minor issue

1. It is mentioned that screening might be rarely used (<10% uptake) in the province. Is the uptake equally low in the different districts and hasn`t increased significantly over time? Incidental diagnosis due to screening may alter the trend results significantly.

Response:

Thank you for your kind remarks. The study we cited found that the national coverage for breast and cervical cancer screening was 9.8%. This rate was not equally low and differed among provinces (1% to 31%). The study also observed that the national breast and cervical cancer screening rate has increased over time, with annual changes of 130% on average.

2. On page 11, it is stated that “the Yogyakarta province, …, has more than twice the national cancer prevalence [2], “Why prevalence? About 200tsd. women living with BC in Indonesia that have been diagnose the past 5 years (https://gco.iarc.fr/today/data/factsheets/populations/360-indonesia-fact-sheets.pdf). Does the prevalence refer to that number? Unfortunately, the resources are in Indonesian only, otherwise I would have checked myself.

“…with BC incidence ranked first in this region [3].” This seems to be a number from the hospital-based cancer registry. Please help the reader to understand the different sources and make sure to be clear about incidence and prevalence.

Response:

Many thanks for this feedback. We cited the prevalence data of cancer from national health survey (reference number 2). This information did not refer to the Globocan estimation since the national health survey data pointed out the cancer prevalence of each province while the Globocan data referred to the national cancer incidence estimation. We used this information regarding the survey statement that the Yogyakarta province has more than twice the national cancer prevalence to underline the importance of conducting a cancer epidemiology study in our province. 

The reference in the sentence “…with BC incidence ranked first in this region [3].” actually meant to refer to Yogyakarta population-based cancer registry data and not only from the hospital-based cancer registry. We apologize for the mistakes in establishing reference no 3 and we have corrected the reference used in this sentence. We also adjusted the manuscript in lines 65-68 to help the readers understand the different sources used.

3. Please state, how was stage collected (UICC/TNM classification?). It is mentioned on page 17, lines 223 ff that “information on the stage at diagnosis was limited, with the percentage of missing data ranging from 73.55%-77.61% in each district”. It might be of value to use essential elements of TNM instead, (e.g. provide proportions of T-stages only).

Response:

Thank you for your kind remarks. The stage information was taken from the clinician's statement of AJCC staging of the patients. If the AJCC staging classification was missing from the data and TNM classification was available, we interpreted the TNM stage into AJCC staging data. However, if both were missing from the data, we considered the stage data missing. In future studies, we will utilize the TNM staging classification.

4. Please check the following links in the references http://labdata.litbang.kemkes.go.id/images/download/laporan/RKD/2018/Laporan_Nasional “An error has occurred. The requested page can't be found.”

Response:

Thank you very much for your suggestion. We apologize for the error in the reference mentioned. We have updated the reference in the manuscript (reference number 2).

Reviewer#2

1. As far as I know, Indonesian population is ethnically complex and surely strongly structured. The population is described in lines 99-106, the database in 126-133 and the sample under study (variables considered) in lines 136-137. but solely as inhabitants of the districts studied. Although the database comprises ethnicity as a variable, why doesn’t the present study consider a ethnic approach, given the importance of genetics in BC?

Response:

Thank you for raising this issue. As part of the limitation of PBCR data that largely relies on the availability of information in its data sources, we cannot control the completeness and variation in data available in each source. This also applies to ethnicity information in the database; the recording of ethnicity was not uniform and was missing in more than 93% of the case in the PBCR database. In addition, we also did not have access to the civil registration data as a base for confirmation. Therefore, this variable was not included in the analysis of the current study. Nevertheless, thank you for your compelling idea. We hope that in the future we can have questionnaires that can capture this phenomenon and use the data for exploring the genetic role in breast cancer development in our population.

2. Line 77. The geographical distribution of BC should be put in perspective: association does not amount to causal relation (“linked to”). Please, clarify this when you cite references 6, 7 and 8 in lines 77-78. Do these works refer to causality or a simple statistical association?

Response:

Thank you for your kind remarks. We clarify that the study in reference number 6 (currently reference number 8) is a case-control study, reference number 7 (currently reference number 9) is a cross-sectional study, and reference number 8 (currently reference number 10) is a prospective cohort study. All these studies applied multivariable analysis and not just a simple statistical association. Based on the study design and the statistical tests demonstrated, we believe that these studies provided evidence of a causal relationship.

3. In the Statistical Analysis, do the “spatial units” amount to districts or subdistricts? This should be explicit in the text for a better understanding.

Response:

Thank you for your comment. In response to that, we have updated the manuscript to include information regarding the spatial unit as a subdistrict in the lines 180-181 and lines 189-190.

4. In lines 188-189, spdep package and rgeoda package should be cited as R packages for those readers unfamiliar with R.

Response:

Thank you for your comment. In response to that, we have updated the manuscript to include the information that spdep and rgeoda were R packages in lines 204-205.

5. The claim in line 193 “During the twelve years observed, 4,268 cases of BC were recorded” cannot be considered as a result but a description of the sample. In my opinion, it should be better moved to another section where the data sample is described. However, this is only a suggestion and I’d rather the authors decide.

Response:

Thank you for your kind remarks. We agreed that this information is unsuitable in the result section, and we have moved this information into the method section for describing the data in lines 152-153.

6. In line 221 the authors consider the stage distribution of BC cases but as mentioned in lines 136-137, the variables considered for the present study were permanent address, year of diagnosis, and sex. Why this paragraph on stage distribution? Please, clarify this. In my view it would be better to mention in lines 136-137 ALL variables considered and not only the MAIN variables.

Response:

We have edited the manuscript in lines 152-153, not separating the main and other variables and instead stating all of the variables.

7. In figure 1, a small map of Java or Indonesia should help those readers unfamiliar to Asia geography where the area under study is.

Response:

Thank you for your suggestion. We have added the Indonesia and Java Island maps in Figure 1 to enable easier identification of Yogyakarta province. We also adjusted the districts' coloring to match the color palette with other figures and increased the font size of the number to help the reader better view the number of the subdistricts.

8. In my opinion, results of I’s Moran and LISA could be provided as supplementary information online.

Response:

Thank you for your kind remarks. Moran I's statistics and LISA results were one of our main analyses in this study, as many previous studies have also investigated the spatial distribution of cancer using these methods. We would propose that these results be kept in the main results.

9. Districts cited in lines 267-269 should also be cited according to numbers, in order to facilitate the ubication in figure 1. Also, those in lines 272-273.

Response:

Thank you for your comment. We have updated the manuscript in lines 287-289 and lines 293-294, writing the subdistrict order aligned with the numbering in Figure 1.

10. I cannot find the results described in lines 278-281.

Response:

The sentences in lines 278-281 (currently lines 299-304) you mentioned are the figure legends of Figure 6, but probably due to the page length they got separated from the Figure title and this might introduce inconvenience. The results are further explained in lines 266-296. We currently added some details in the figure legend to help readers identified the clustering results in lines 301-302.

11. In the discussion from lines 317-319, it would be perhaps of help for the authors the reference Solikhah et al. (2019) Awareness Level about Breast Cancer Risk Factors, Barriers, Attitude and Breast Cancer Screening among Indonesian Women. Asian Pac J Cancer Prev. 20(3):877-884. doi: 10.31557/APJCP.2019.20.3.877. here it is concluded that “Urban women had a poorer level of knowledge of breast cancer risk factors compared to women living in more rural areas” (!!). It seems that the grade of “awareness” can affect the incidence of BC. This is also valid for line 349 (cite 49), where “awareness” is explicitly mentioned.

Response:

Thank you for your kind suggestions. The paragraph in lines 317-319 you mentioned (currently lines 341-343) explains the incidence of BC and its comparison in developed countries. It is elaborated that although Indonesia’s current incidence is lower than that in developed countries, the rate is predicted to grow rapidly based on the projection studies we cited in reference number 34. We have yet to discuss the BC screening in this section.

In and around the paragraph of line 349 you mentioned (currently line 385), we tried to explain several possible clauses of clustering of BC ASR in the Kota Yogyakarta/Yogyakarta City District. Urbanization, higher human development index (HDI), higher per capita expenditure, and higher education might explain the high BC ASR in the district. The reference for these findings were sporadic studies with various sources. In fact, these potential risk factors of high BC ASR in Yogyakarta City District warrant a further investigation in future studies with appropriate design. These factors mentioned were disparities at the regional level while Sholikhah et al. discussed a different focus namely individual risk factors of BC risk factor knowledge. Furthermore, citation number 49 (currently number 54) observed that higher per capita expenditure and higher education positively correlated with higher screening awareness (mammography).

However, the study by Sholikhah et al. that you mentioned has a very important finding regarding BC screening, and we thank you for pointing out this study. We accommodated the findings in Solikhah et al. in lines 420-422 and reference number 73 to further underline the situation of breast cancer screening in Indonesia. 

12. Line 330. Please, write the numbers of the six districts (names can also be included). Given that much of the discussion can be seen in figure 1-6, as a rule, include indications such as “see figure 3” or so across the whole discussion.

Response:

Thank you for your critical suggestions. As you suggested in the result section, we have numbered the subdistrict referring to Figure 1. Moreover, in line 330 (currently lines 356), we have also included the name and number of subdistricts mentioned in the paragraph, through lines 352-359. Throughout the manuscript, we have added the indicators to refer to the relevant figures/tables and added line 210 to refer to S1 Table and line 268 to refer to S2 Table.

13. This comment is both for “Study strength and limitation” and also for the conclusions. Genetics are just once mentioned in the whole text (line 75). It has been widely reported the importance of genetic polymorphism in BC. Also, genetics is strongly correlated with ethnicity. A reflection on this should be made in order to make the readers aware that the authors consider genetics as a risk factor.

Response:

Thank you for pointing out this matter. We realized that genetic factors are one of the important factors when discussing BC incidence. Although not currently the focus of this study of geographical BC disparities, studies regarding the genetics of breast cancer are one of our future directions. To express to the readers that we acknowledge this issue, we have added our thoughts on addressing genetic factors and the implication of this cluster analysis study to future BC genetic studies in the discussion section lines 367-377.

14. Line 106. It reads Fig 1 but in line 209 Figure 2. ¿Fig or Figure?

Response:

Thank you for your critical observation. We realized the inconsistency and updated the manuscript to address this issue in line 214.

15. Not everyone knows of Indonesian geography. The authors refer sometimes “Yogyakarta province” and sometimes only “Yogyakarta”. Fig 1 seems to suggest that Yogyakarta is a city and Wikipedia cites Yogyakarta as the “capital city” of the region of Yogyakarta. For the European context, a “province” is a territory where several cities can be located and districts use to be only within the cities. Perhaps it is obvious for the authors, but a clarification across the whole text would help many readers.

Response:

Thank you for your important remarks. To helped the readers understand and emphasize the difference, we changed the term "Kota Yogyakarta" to "Yogyakarta City" throughout the manuscript. We added the explanation that "Yogyakarta City" is also the capital city of "Yogyakarta Province" in lines 102-103 to underline the administrative boundaries in a better way. We also made adjustments in the figures (Fig 1-5) accordingly.

16. Since I searched in pubmed the terms “Indonesia” and “breast cancer”, I found several references of the years 2020-2023 that I am not sure why the authors do not even mention. Please, check this mainly in the introduction.

Response:

Thank you for your kind remarks. While drafting the manuscript, we have also searched for studies conducted recently between 2020-2023. We previously have listed relevant recent breast cancer studies in Indonesia from 2020-2023 in references 18 (regarding BC geographical disparities) and 69 (regarding BC screening rate). Acknowledging your remarks, we have included the 2021 study by Nindrea et al. for the reference number 7 in lines 75-76, which examines the reproductive, high-fat diet, and BMI risk factors for breast cancer. Although not in the 2020-2023 period, we also included the 2000 study by Wakai et al., also for the reference number 6 lines 75-76. This case-control study explores the association between fat intake and breast cancer in Indonesia. We also included earlier studies in 2019 by Solikhah et al. in reference number 73 regarding BC screening awareness as suggested in your previous review number 11.

We hope that, following our revisions, the manuscript is now suitable for publication in the PLOS ONE. 

Yours sincerely,

Susanna Hutajulu, MD, PhD

---

## [Decision Letter · Decision Letter 1]

19 Jun 2023

Breast cancer incidence in Yogyakarta, Indonesia from 2008-2019: a cross-sectional study using trend analysis and geographical information system

PONE-D-23-05461R1

Dear Dr. Hutajulu,

We’re pleased to inform you that your manuscript has been judged scientifically suitable for publication and will be formally accepted for publication once it meets all outstanding technical requirements.

Kind regards,

Abdulkader Murad, Ph.D

Academic Editor

PLOS ONE

Additional Editor Comments (optional):

Reviewers' comments:

Reviewer's Responses to Questions

**Comments to the Author**

1. If the authors have adequately addressed your comments raised in a previous round of review and you feel that this manuscript is now acceptable for publication, you may indicate that here to bypass the “Comments to the Author” section, enter your conflict of interest statement in the “Confidential to Editor” section, and submit your "Accept" recommendation.

Reviewer #1: All comments have been addressed

Reviewer #2: All comments have been addressed

2. Is the manuscript technically sound, and do the data support the conclusions?

Reviewer #1: Yes

Reviewer #2: Yes

3. Has the statistical analysis been performed appropriately and rigorously? 

Reviewer #1: Yes

Reviewer #2: Yes

4. Have the authors made all data underlying the findings in their manuscript fully available?

Reviewer #1: No

Reviewer #2: Yes

5. Is the manuscript presented in an intelligible fashion and written in standard English?

Reviewer #1: Yes

Reviewer #2: Yes

6. Review Comments to the Author

Reviewer #1: All questions were adressed, is a sound pieice of scientific work, interesting to read and important fpr preventive efforts. Thank you for the profound explanations and revisions.

Reviewer #2: The authors have fulfilled all the requirements by this reviewer. Now the manuscript is much more improved.

7. PLOS authors have the option to publish the peer review history of their article (what does this mean?). If published, this will include your full peer review and any attached files.

Reviewer #1: No

Reviewer #2: No

---

## [Editor Report · Acceptance letter]

26 Jun 2023

PONE-D-23-05461R1 

Breast cancer incidence in Yogyakarta, Indonesia from 2008-2019: a cross-sectional study using trend analysis and geographical information system 

Dear Dr. Hutajulu:

I'm pleased to inform you that your manuscript has been deemed suitable for publication in PLOS ONE. Congratulations! Your manuscript is now with our production department. 

Kind regards, 

on behalf of

Professor Abdulkader Murad 

Academic Editor

PLOS ONE